# Decision Support for Landscapes with High Fire Hazard and Competing Values at Risk: The Upper Wenatchee Pilot Project

**Haley K. Skinner [1,\*], Susan J. Prichard [2]** and **Alison C. Cullen [1]**

[1] Evans School of Public Policy and Governance, University of Washington, Seattle, WA 98195, USA; alison@uw.edu

[2] School of Environmental and Forest Sciences, University of Washington, Seattle, WA 98195, USA; sprich@uw.edu

\* Correspondence: hkskinner97@gmail.com; Tel.: +1-(570)-396-4076

**Abstract:** Background: Climate change is a strong contributing factor in the lengthening and intensification of wildfire seasons, with warmer and often drier conditions associated with increasingly severe impacts. Land managers are faced with challenging decisions about how to manage forests, minimize risk of extreme wildfire, and balance competing values at risk, including communities, habitat, air quality, surface drinking water, recreation, and infrastructure. Aims: We propose that land managers use decision analytic frameworks to complement existing decision support systems such as the Interagency Fuel Treatment Decision Support System. Methods: We apply this approach to a fire-prone landscape in eastern Washington State under two proposed landscape treatment alternatives. Through stakeholder engagement, a quantitative wildfire risk assessment, and translating results into probabilistic descriptions of wildfire occurrence (burn probability) and intensity (conditional flame length), we construct a decision tree to explicitly evaluate tradeoffs of treatment alternative outcomes. Key Results: We find that while there are slightly more effective localized benefits for treatments involving thinning and prescribed burning, neither of the UWPP's proposed alternatives are more likely to meaningfully minimize the risk of wildfire impacts at the landscape level. Conclusions: This case study demonstrates that a quantitatively informed decision analytic framework can improve land managers' ability to effectively and explicitly evaluate tradeoffs between treatment alternatives.

**Keywords:** climate change; fuel treatments; fire management; risk

## 1. Introduction

Climate change is driving rapid changes to western North American (wNA) forest ecosystems. With warmer, drier, and lengthening wildfire seasons, large wildfires are increasingly common and severe [1–6]. A legacy of fire exclusion in many fire-prone forests due to settler colonialism and fire suppression [7] has caused forests to become denser and wildland fuels to accumulate with concomitant changes in fire regimes and forest resilience [8,9].

The return of fires to many wNA forests has been accompanied by increasingly severe impacts to forest ecosystems [10,11] and communities [12,13]. Land managers are tasked with challenging decisions about how to manage forests and intensifying wildfire risk. These decisions require careful consideration of several competing objectives. For example, as the wildland urban interface (WUI) expands with more people living within or close to fire-prone forests, fuel reduction treatments that involve prescribed burning may be challenging to implement due to perceived risks and potential smoke impacts. However, wildfire risk reduction is critically needed to mitigate future wildfire impacts to WUI communities [14].

Several tools exist to assist decision making for wildfire management, though some are less accessible than others to decision makers. Quantitative wildfire risk assessments (QWRAs) offer an integrated raster-based mapping of potential risk across an entire landscape [15,16]. As a complement to QWRAs, decision analytic frameworks—such as decision trees—allow risk managers to explore the available options and to confront the scientific, social, and political uncertainties that result in tradeoffs in the prioritization of those options, by estimating probabilities for a range of possible outcomes, represented as the right-most nodes on a decision tree [4,17–22]. Decision frameworks are thus used to assess overall risk mitigation associated with available risk management strategies and their likely consequences [4,17–22]. This paper presents a case study to offer an improved methodology for utilizing the Interagency Fuel Treatment Decision Support System (IFTDSS) in conjunction with a decision analytic approach that elevates the quality of risk analysis and the risk management process and offers comprehensive insight into the tradeoffs associated with different fuel treatment options.

## 2. Background

In the decades following the introduction of prescribed fire use as a means of proactively minimizing risk of intense wildfire, several approaches for wildland fuels management have been introduced to the portfolio of options available to managers. These approaches fall into two main categories: fuel reduction and fuel rearrangement. Fuel reduction treatments include prescribed fire, thinning, or other mechanical treatments followed by the removal of any remaining debris (either burning or removal from site). Fuel rearrangement includes thinning or mechanical treatment (also known as mastication) where debris is left on site [23–25]. Effective fuel reduction, including prescribed burning and combinations of thinning from below followed by prescribed burning, promotes lower flame length fires, reduces tendency to spread, and decreases likelihood of crown fire [25–27].

Cultivating fire-resilient landscapes requires informed, strategic planning and knowledge of where individual treatments will be most effective and how they may mitigate fire impacts across landscapes [25,28,29]. It is often the case that one treatment approach alone is insufficient to effectively mitigate extreme fire risk. For example, in some cases, thinning alone can effectively reduce risk of severe fire, but it is important to acknowledge that without some means of removal of residual debris (slash), thinning can also contribute to fire risk [25]. Similarly, in many fire-excluded forests, prescribed burning alone may not achieve objectives of reducing forest densities because too many trees would survive the burning treatment [27]. Thus, managers often opt to pair thinning with prescribed fire [24,26]. Still, thinning and burning are not always appropriate for a given forest structure or landscape; for example, a montane or subalpine forest dominated by thin-barked trees would not be a good candidate for thinning and prescribed burning [25]. Herein lies a challenge for fire managers: where should managers administer fuel treatment and which specific approach should be used?

### 2.1. Uncertainty and Existing Decision Support

Decisions surrounding fuel treatment are constrained by uncertainty and variability in fire ignition, weather, location of occurrence, and many other factors [30–32]. Several tools exist to serve decision makers at different points in the process of wildfire planning and management. Among these tools are fuel and fire weather information, the Interagency Fuel Treatment Decision Support System (IFTDSS) and the Wildland Fire Decision Support System (WFDSS). As detailed in Table 1, each of these tools has value for wildland fire decision making with some overlap in functionality.

**Table 1.** Overview of existing key decision support tools.

| Tool | When to Use | Useful For |
|---|---|---|
| IFTDSS | In advance of fire. | Proactive fuel planning, fire behavior modeling, analyzing risk of fire on treated/untreated land [33]. |
| Fuel and Fire Weather Information (i.e., NFDRS) | During initial attack. | Using weather and climatology data to estimate fire danger and inform initial attack decisions; when there is not enough time to execute full analysis in WFDSS [34–36]. |
| WFDSS | During active fire, after initial attack. | Active fire management, documentation of fire status, characteristics and firefighting strategies, strategizing resource deployment and firefighting tactics, fire behavior modeling, analyzing risk of fire on landscape as is [34,37–39]. |

IFTDSS is a webtool that provides relatively user-friendly access to fire behavior modeling capabilities and geospatial data from several sources including LANDFIRE, SILVIS, and various federal agencies with the option for users to add their own data [40–42]. The main purpose of IFTDSS is to facilitate effective proactive fuel treatment planning. It allows users to examine the response of an area of interest to expected fire behavior under a wide variety of weather conditions. Users can easily generate maps, graphs, and tables that describe the impact that fire is expected to have on a variety of highly valued resources and assets (HVRAs) on the landscape, such as air quality, water quality, and habitat.

*2.2. Decision Analytic Frameworks as a Complementary Decision Support Tool*

A key gap in the existing tools available to wildland fuel managers is noted in Rapp et al. (2020) [34]: even when using decision support tools such as WFDSS or IFTDSS as intended, decision makers are still challenged to weigh tradeoffs and ultimately make decisions based on their own expertise and judgment and any relevant external factors, in the face of substantial uncertainty. The planning process can be challenging because land managers are often specialized in their respective disciplines and the many explicit tradeoffs may not be within their scope of their work.

Decision making and planning often requires environmental analyses such as Environmental Impact Statements (EIS), which are generally written with distinct and sometimes siloed sections dedicated to project impacts to fish, forests, and habitat. While the EIS attempts to integrate these impacts, in reality it is very difficult to consolidate these risks into a unified whole [43,44]. Decision making is further challenged by competing concerns about retaining LSR habitats for Northern Spotted Owl (NSO; *Strix occidentalis caurina*) and other wildlife species dependent on old forests and the risk of stand-replacing wildfire events [45,46]. Habitats in fire-prone landscapes—especially in LSRs, as they have been identified in the Northwest Forest Plan (NWFP)—are characterized by old forest structures with multilayered canopy layers and abundant coarse wood. Overharvesting in the 20th century led to steep reductions in old forests. The NWFP established LSRs in northwestern California, western Oregon, and western Washington and designated these as moist forests. The NWFP area also spans over the Cascade Crest into eastern Oregon and Washington State into dry forests. Although language within the NWFP allowed for fire hazard mitigation within late successional reserves in dry forests, they have generally been treated as set asides with no active management [47]. The Upper Wenatchee Pilot Project (UWPP) contains LSRs that represent tradeoffs between continued protection of late successional habitat and addressing threats from large, high-severity wildfires. Competing perspectives can intensify the difficulty of making such a decision around values at risk [48].

An advantage of IFTDSS as a decision support tool is that it offers a relatively accessible and integrated way to examine potential fire behavior, given a particular fuel model or treatment scenario, and to conduct a QWRA on identified values at risk. A QWRA is a method for understanding potential impacts of wildfire on a landscape that yield a geospatial analysis of potential benefits and threats, given specified fire weather conditions

represented as both a conditional weighted net value change (i.e., conditional on the fact that a fire has occurred) and an expected weighted net value change for each pixel on the landscape (see Figure 1) [15,49,50]. Decision analytic frameworks—such as decision trees—when combined with this methodology go beyond the QWRA process to offer a broad, systems view of tradeoffs that can complement existing tools like IFTDSS and may offer a more intuitive lens through which to analyze IFTDSS outputs. For this reason, decision analytic approaches are well suited for comparing the consequences of prioritizing one value/asset over another.

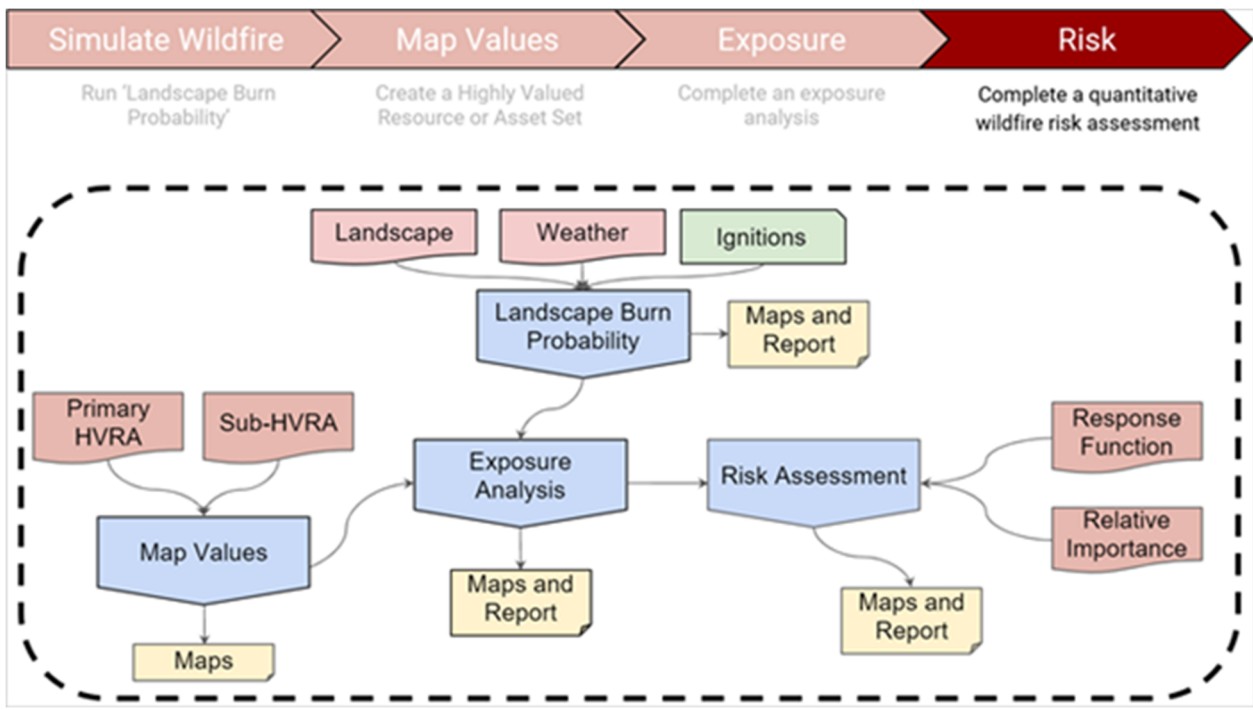

**Figure 1.** Inputs and outputs of wildfire simulation to assess risk. Required user-entered inputs are represented by red, while processes are represented in blue and outputs in yellow (landscape, weather, primary HVRA, sub-HVRA, response function, and relative importance) [40].

Decision analytic frameworks have been applied to improve and analyze challenging environmental policy decisions and their relevant tradeoffs, ranging from climate change adaptation measures [51] to risk management of contaminated industrial sites [52]. Various decision analytic frameworks have been suggested and applied to forest and wildfire management, tradeoffs, and the social costs thereof [14,17–20]. Similarly, Hirsch et al. (1979) [18], Radloff et al. (1982) [19], and Cohan et al. (1984) [30] developed early applications of decision analysis techniques and fire modeling to decisions surrounding thinning/slash and removal and prescribed fire usage, respectively. Similarly, McGregor et al. (2017) [53] introduced a simple decision tree framework which considers the risk of wildfire given pre-existing fuel conditions to determine whether to suppress a fire or let it burn. These studies, however, tend to focus on probabilistic outcomes in terms of fire intensity or burned area across a landscape. Given the increasing challenges and uncertainties associated with wildfire management, there is a need for decision support that optimizes efficient and effective fuel treatments and long-term reduction of risk and improves weighing of tradeoffs [54]. We address this need by utilizing existing decision support tools (IFTDSS and QWRAs) in conjunction with a decision analytic framework, which can not only minimize fire intensity across the landscape, but also estimate probabilistic impacts to HVRAs.

## 3. Materials and Methods

Our novel approach to improving decision making using a decision analytic framework contains several steps, which we discuss below and are outlined in Figure 2.

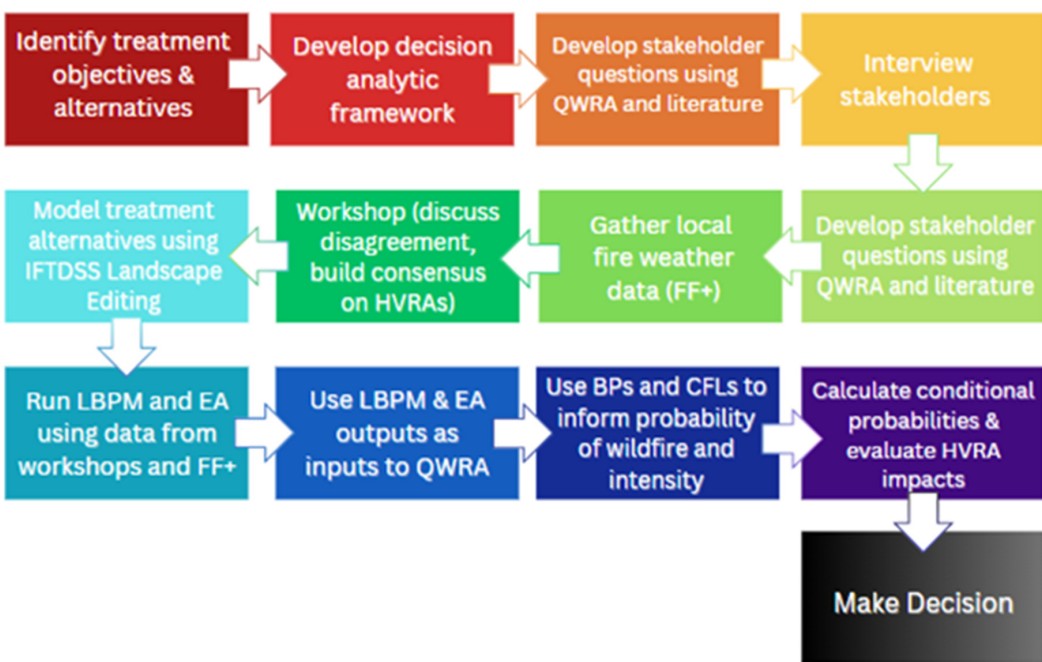

**Figure 2.** Process flow for improved decision making using a decision analytic framework and QWRA. This figure displays the step-by-step process demonstrated in this paper through the UWPP case study.

### 3.1. Study Area

We selected the UWPP area in eastern Washington State because it is the focus of planning to address high fire risk, and it is characterized by competing values at risk. This area is densely forested and comprises 30% congressional reserved areas, another 30% of which is designated as late successional reserves (LSR) [55]. Existing wildfire risk in this area poses a direct threat to old and mature forests, habitat of the threatened northern spotted owl and several threatened anadromous fish species. This is of particular concern given the land allocations required by the Northwest Forest Plan, which designate 38% of the planning area as LSRs and 14% of the area as riparian reserves [56]. These designations require the following:

(1)  Any restoration treatment on LSR needs to protect and enhance the late-successional and old-growth forest ecosystems;

(2)  Prohibition and regulation of activities that will inhibit the maintenance and restoration of species composition and structural diversity of plant communities in riparian reserves [56].

Wildfire also threatens nearby communities in the WUI, including Plain, Chumstick, and Leavenworth.

Fire risk to these communities continues to increase with climate change to such an extent that insurance companies are not only increasing rates but they have already begun refusing coverage to homeowners [57,58]. While the region surrounding the UWPP has a history of large fires, including three that burned nearly 17,000 acres (68.8 square kilometers) total during the summer of 2022 when this research was being conducted [56,59], the UWPP landscape has experienced very little stand-replacing wildfire in recent years [60]. Thus, this high-risk landscape is especially ripe for devastating wildfire, and, as a result, is the focus of the collaborative UWPP, led by the USFS, which seeks to restore forest health, reduce

wildfire risk, and improve wildlife habitat and watershed function, further confirming the selection of this area for our study.

The UWPP team has examined potential proactive treatment options, including thinning, prescribed fire, a combination of the two, stand regeneration, and options that prioritize NSO habitat over ecological resilience [56]. Within the UWPP area, USFS has proposed two alternative treatment scenarios (mapped in Figure 3). Alternative 1 (left) represents a prioritization of fuel treatment actions such as thinning and prescribed burning to reduce fire risk, while Alternative 2 (right) reflects a prioritization of the threatened NSO [56]. The location of treatment areas in both alternatives was centralized to be close to communities for WUI protection and by existing road networks and priority habitat for NSO.

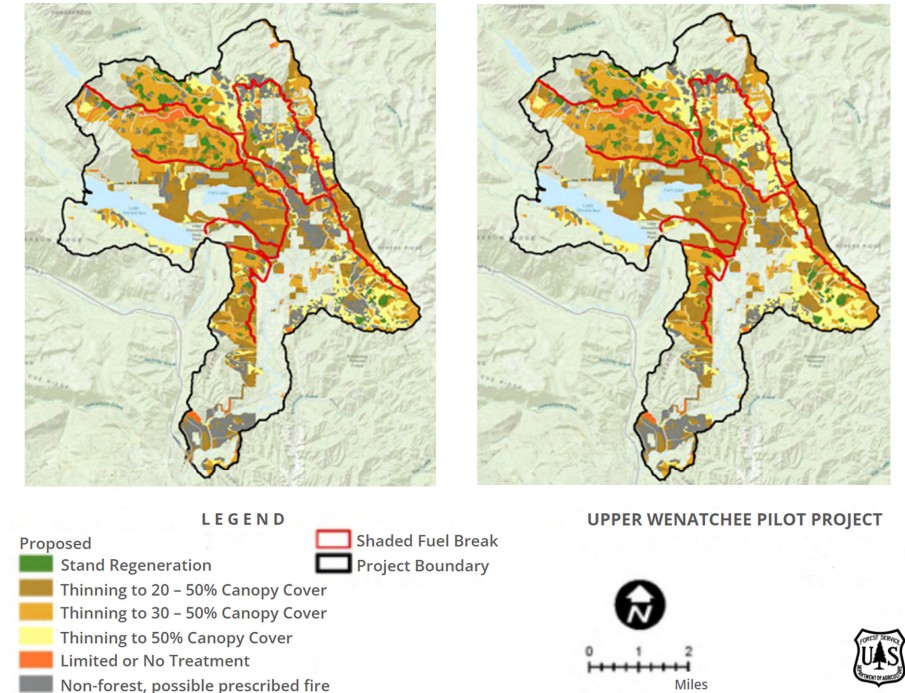

**Figure 3.** Comparison of alternatives under consideration for the UWPP. **Left panel**: Alternative 1 prioritizes pretreatment. **Right panel**: Alternative 2 prioritizes habitat for the northern spotted owl [56].

### 3.2. Expert Interviews and Workshops

Identification and analysis of potential impacts of wildfire HVRAs to each unique landscape rely heavily on the input of stakeholders with localized expertise. Thus, the majority of our data collection occurred through two approaches: a series of individual semistructured interviews with local experts and a virtual stakeholder workshop during which participants discussed their unique perspectives and informed an integrated assessment of values at risk. We interviewed 10 individuals with expertise varying from wildlife biology to silviculture, forestry, fire ecology, landscape ecology, and aquatic ecology (see Appendix A). We selected participants due to the applicability and diversity in profession and their experience working directly or indirectly with the UWPP landscape and planning processes. Seven of our ten original interviewees also participated in our virtual workshop.

During interviews, participants identified a set of HVRAs and sub-HVRAs relevant to UWPP. The subsequent workshop included discussion of the assignment of relative importance (RI) for these HVRAs. RI is a numeric weight which reflects the need for tradeoffs and prioritization of some HVRAs over others, whether because of policy, land management objectives, or other competing interests [50]. Participants were asked to assign

response functions (RFs), which are numeric scores from −100 to 100 representing the susceptibility or response of an HVRA to wildfire at different levels of intensity based on first order fire effects, where −100 is extreme loss and 100 is extreme benefit [50].

At the subsequent workshop, we first introduced default HVRAs within IFTDSS. We then presented participants with visual analyses of their responses and asked them to elaborate on points of dissent, outlying responses, and individual perspectives. The presence of specialists from multiple fields allowed the group to develop mutual clarity on the issues and promoted a more unified set of responses than the individual interviews. The stakeholder engagement process contributed several sources of guiding information. First, interviews confirmed that FireFamilyPlus information used as inputs to the IFTDSS Landscape Burn Probability Model (LBPM) are appropriate for the landscape (such as fuel moistures, probability of spotting, wind speeds, etc.). Second, they ensured that all HVRAs of concern are represented appropriately. Third, our workshop findings allowed us to integrate localized expert opinion on the RI and RF of each key value (HVRA) of the UWPP landscape. As a result of the workshop, we determine the set of HVRAs of greatest concern on the UWPP landscape.

### 3.3. Decision Analytic Frameworks

To support pretreatment-related decision making, we developed a decision tree template to represent wildfire management approaches and outcomes based on the probability of wildfire severity under 90th and 99th percentile weather (Figure 4). In this case study, we focused on the portion of the decision pertaining to combined mechanical thinning and fire use (dark green). We modeled the decision that land managers currently face in choosing between treatment alternatives under consideration in the UWPP (Figure 5). We used the Landscape Editing function in IFTDSS to model and evaluate several treatment scenarios. The Landscape Editing function allowed us to specify different levels of thinning and prescribed burning to accurately represent UWPP treatment alternatives (see Table 2).

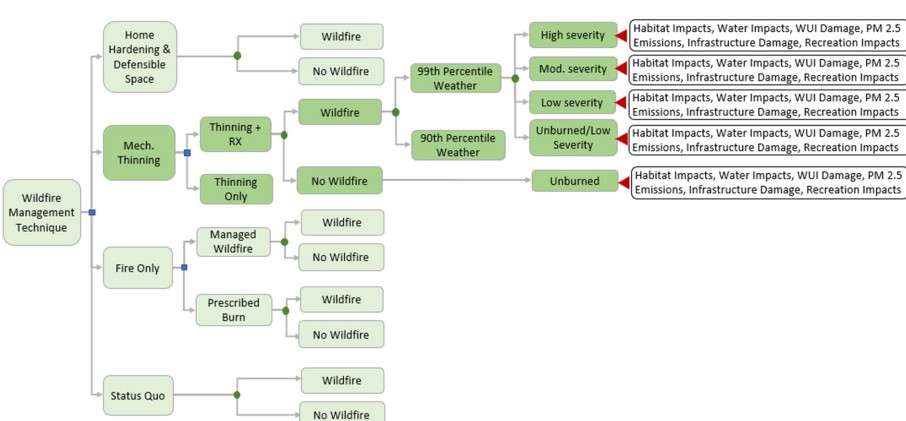

**Figure 4.** Wildfire management technique decision tree. This decision tree diagram is designed to be read from left to right. A blue square represents a "decision node" (a point where the decision maker must choose between multiple options). A green circle represents an "uncertainty node", where uncertainty about future outcomes enters the equation. A red triangle represents a "terminal node", the point at which all impacts and consequences of a possible future outcome accrue to the decision maker and stakeholders. Dark green squares represent uncertainties which can be informed using values generated by IFTDSS. Branches that end without a terminal node are assumed to continue along a similar path to analogous branches in the tree, a shorthand used here to save space. For example, as highlighted by the dark green pathway, a decision maker may choose to mechanically thin and to further pair thinning with prescribed burning (RX). All boxes to the right of those two decision nodes—"Wildfire" vs. "No Wildfire", "99th Percentile Weather" vs. "90th Percentile Weather", and high severity, moderate severity ("Mod."), unburned and/or low severity of fire—represent uncertain but possible future outcomes of the decisions made.

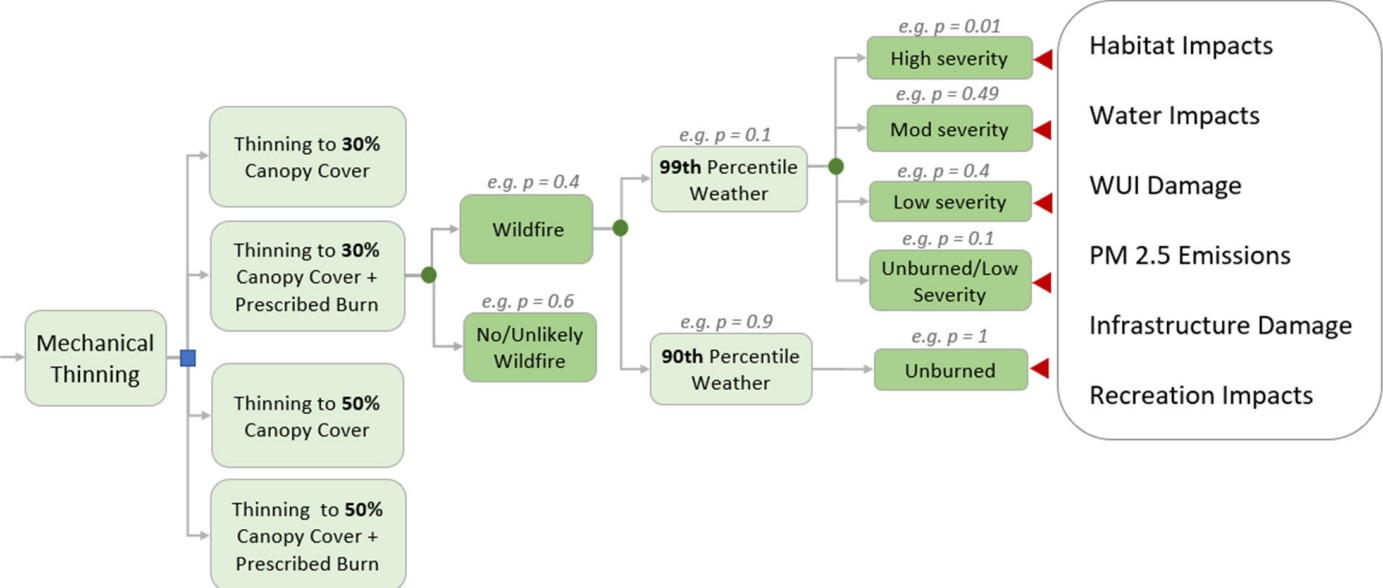

**Figure 5.** Decision analytic representation of tradeoffs associated with mechanical thinning strategy. This decision analytic framework represents the uncertainties and potential outcomes surrounding the decision about treatment of a 30 m pixel within the landscape of interest. In this figure, we trace a branch within the dark green pathway from Figure 3 and proceed with the assumption that the decision maker has chosen to mechanically thin a landscape. At the next decision node (moving to the right), the decision maker must choose between (1) thinning to 30% canopy cover, (2) thinning to 30% canopy cover followed by prescribed burning, (3) thinning to 50% canopy cover, and (4) thinning to 50% canopy cover followed by prescribed burning. From there, the decision maker confronts the uncertainty about whether or not a wildfire ignition will occur and the probabilities thereof followed by the uncertainty of extreme fire weather conditions, which are represented probabilistically. Each branch terminates with the realization of all outcomes and consequences associated with this specific sequence of decisions and uncertainties.

**Table 2.** Landscape editing rules. For each UWPP alternative, we edited landscapes to reflect each treatment alternative. These edits were applied according to the proposed geospatial range for thinning treatment according to each alternative.

| Treatment Scenario | Apply Where... | Canopy Cover | Canopy Bulk Density (kg/m$^3$) | Fuel Model | Canopy Base Height (m) |
|---|---|---|---|---|---|
| Thinning to 30% Canopy Cover | Canopy Cover > 30% | 30% | 0.07 | 164 (TU4) | 10 |
| Thinning to 30% Canopy Cover + Prescribed Burn | Canopy Cover > 30% | 30% | 0.07 | 101 (GR1) | 10 |
| Thinning to 50% Canopy Cover | Canopy Cover > 50% | 50% | 0.15 | 164 (TU4) | 10 |
| Thinning to 50% Canopy Cover + Prescribed Burn | Canopy Cover > 50% | 50% | 0.15 | 101 (GR1) | 10 |

We designed a decision tree capable of representing a wide range of proactive strategies that planners can pursue to minimize the risk of severe wildfire impacts. For our application, the decision tree displays each treatment strategy individually, while in practice, multiple treatments are typically combined [24,26]. Still, the tree structure is valuable in presenting a comprehensive and realistic view of the tradeoffs and probabilistic outcomes associated with various treatment alternatives which have implications for potentially competing values at risk.

To illustrate the application of this decision analytic approach, we simplified the analysis to include only the first two treatment scenarios displayed in Figure 5: thinning to 30% canopy cover and thinning to 30% canopy cover followed by prescribed burning. We considered both scenarios for each treatment alternative for the UWPP.

### 3.4. Fire Weather Data

As reflected in the fourth column of Figure 5, we conducted landscape burn probability modeling in IFTDSS for two fire weather conditions: 90th percentile fire weather and 99th percentile fire weather. To model these fire weather scenarios, we input weather conditions such as wind speed and direction, fuel moisture, spotting probability using FireFamilyPlus, a software package that uses hourly fire weather observations collected from RAWS (that is, Remote Automated Weather Stations) to calculate fuel moistures and indices from NFDRS (Missoula Fire Sciences Laboratory). Appendix B offers an example of data collected.

### 3.5. Deriving Probabilities

To derive probabilities associated with each uncertainty specified in Figure 5, we simulated wildfire occurrence subject to specified weather conditions. The LBPM capabilities of IFTDSS allows the quantification of several outputs, including conditional burn probability and conditional flame length. First, we used Table 3 to construct conditional flame length classes akin to those used in IFTDSS.

**Table 3.** Relationship of surface fire flame length to fire intensity. Source: Andrews and Rothermel (1982) [61].

| Flame Length | Fireline Intensity | Interpretation |
|:---:|:---:|:---|
| *Feet*<br><4 | *Btu/ft/s*<br><100 | Fire can generally be attacked at the head or flanks by persons using handtools.<br>Handline should hold the fire. |
| 4–8 | 100–500 | Fires are too intense for direct attack on the head by persons using handtools.<br>Handline cannot be relied on to hold fire.<br>Equipment such as plows, dozers, pumpers, and retardant aircraft can be effective. |
| 8–11 | 500–1000 | Fires may present serious control problems – torching out, crowning, and spotting.<br>Control efforts at the fire head will probably be ineffective. |
| >11 | >1000 | Crowning, spotting, and major fire runs are probable.<br>Control efforts at head of fire are ineffective. |

We then estimated the probability of wildfire in Figure 5 using both geospatial and statistical analyses of burn probability outputs. Specifically, we interpreted the burn probability percentages presented in Figure 6 as the probability that a pixel in this space would experience each burn probability or conditional flame length, conditional on all combinations of decision alternatives and chance events that precede these outcomes. Thus, the probability of wildfire is constructed as the weighted sum of the probabilities of low, moderate, and high fire severity in the landscape.

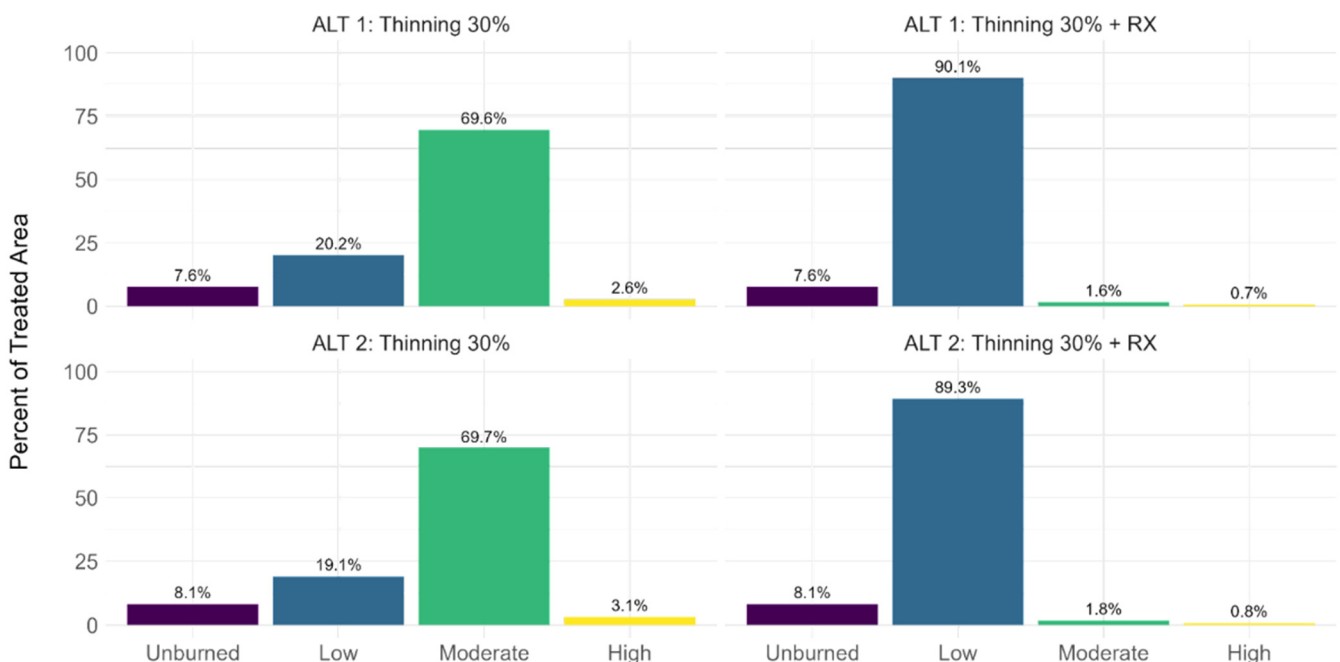

**Figure 6.** Fire severity distributions based on conditional flame length under 99th percentile fire Weather. "Unburned" implies the share of landscape that was burnable but did not burn, "Low" = 0–4 feet (0–1.2 m), "Moderate" = 4–8 feet (1.2–2.4 m), and "High" = 8–12+ feet (2.4–3.7+ m).

## 4. Results

### *4.1. Workshop Results and Prioritizing HVRAs*

During our workshop, stakeholder participants were asked to review a set of default HVRAs included in IFTDSS. These stakeholders generally agreed that the default HVRAs were important for the UWPP and further identified several sub-HVRAs to be added for explicit consideration. Notably, stakeholders recommended that the term Habitat refer specifically to that of NSOs, fish, old growth forests, white-headed woodpeckers (*Picoides albolarvatus*), nonforest, and other early successional habitat (see Appendix C for further discussion of HVRA selection and data). The workshop yielded consensus in terms of relative importance and response functions, though not without lengthy discussion (Table 4). Experts were asked to identify a "most important" HVRA; however, they exhibited a tendency to rank all HVRAs highly. When pressed to be decisive about their top priority, experts identified People and Property as the most important value in the UWPP landscape, followed by Habitat and Infrastructure.

**Table 4.** Primary HVRA RI refers to the relative importance score assigned to each HVRA by stakeholders.

| PRIMARY HVRA: | RI (0–100) |
|---|---|
| Habitat | 90 |
| People and Property | 100 |
| Air Quality and Emissions | 60 |
| Recreation | 50 |
| Surface Drinking Water | 70 |
| Infrastructure | 90 |

## 4.2. Informing Uncertainty

Across all scenarios, conditional burn probability is generally low across the majority of the UWPP study area and designated treatment areas relative to the surrounding region. Figure 6 shows a sample analysis of UWPP Alternative 1 subject to 99th percentile fire weather. The panels (moving from left to right) reflect burn probability (BP) without treatment, burn probability after thinning to 30% canopy cover, and burn probability after thinning to 30% canopy. This figure and the top panels of the corresponding histogram in Appendix D reveal only modest differences in burn probability across treatment scenarios within UWPP Alternative 1. Appendix E offers a similar histogram analysis by forest type.

Although burn probability is generally quite low across the study area (Figure 7 and Appendix D), if a wildfire did occur, it would be expected to result in high flame lengths under both the 90th and 99th percentile weather scenarios (Figure 8). Areas treated with combined thinning and prescribed fire exhibit much lower expected conditional flame length than those associated with either thinning alone or the no-treatment scenario. Lower flame lengths are also concentrated in and around the treated area.

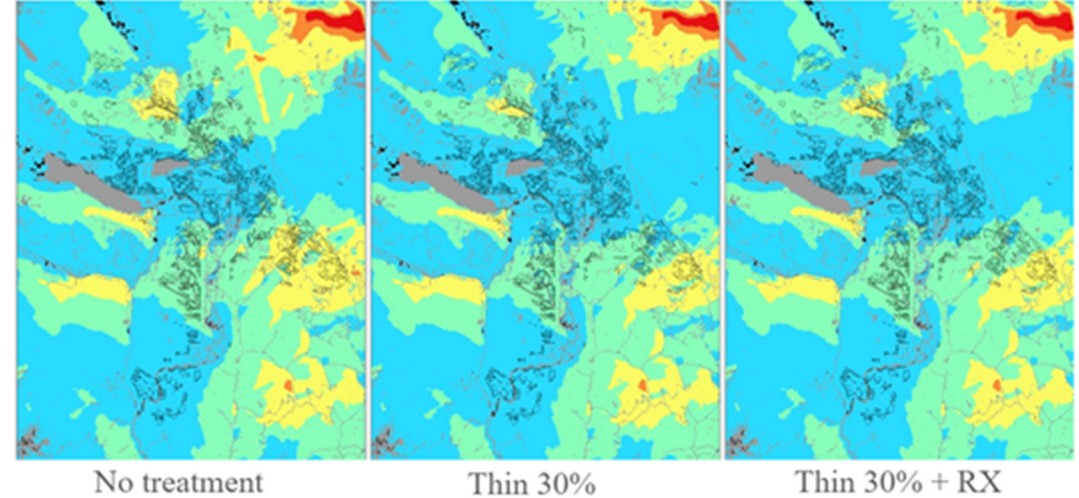

**Figure 7.** Comparison of conditional burn probabilities by treatment approach. Conditional burn probabilities for UWPP Alternative 1 subject to 99th percentile fire weather are depicted for three treatment scenarios: no treatment, thinning to 30% canopy cover, and thinning to 30% canopy cover followed by prescribed burning. The treatment area is overlaid as a black outline.

As reflected in Figure 6, thinning alone yielded similar expected burn probabilities across each alternative (i.e., the majority of the treated area experiences low burn probability). Across both alternatives, we found that the probability of moderate flame lengths is highest when conditioned by the "Thinning 30%" treatment scenario, whereas the probability of low flame lengths is highest for combined thinning and prescribed burn.

Based on these results, Figure 9 represents a landscape that has been thinned to 30% canopy cover and burned. Assuming 99th percentile weather, there is a 92.4% chance that wildfire will occur in a given pixel and about a 90.1% chance that it will be a low-severity fire. At the terminal node at the far right of the decision tree, we observe a 0.06% probability that there is a high-severity fire with all the subsequent impacts accruing, conditional on a wildfire occurring in 99th percentile weather after thinning to 30% canopy cover combined with prescribed burn. Appendix F presents probability calculations for additional intermediate probabilities not evaluated in this figure.

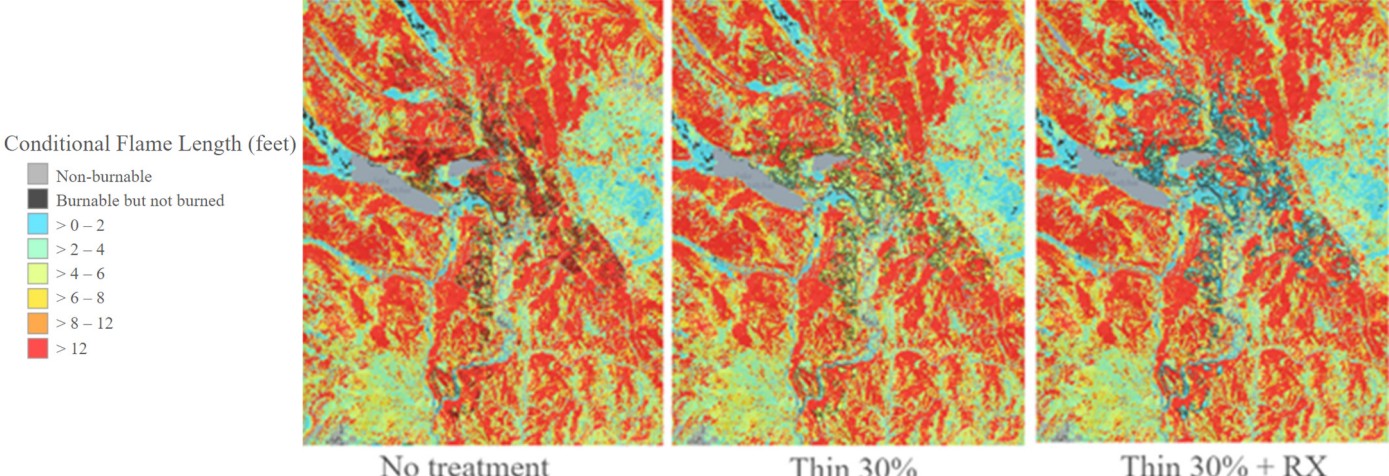

**Figure 8.** Comparison of conditional flame lengths by treatment approach. Conditional flame lengths for three treatment scenarios for UWPP Alternative 1 subject to 99th percentile fire weather: no treatment, thinning to 30% canopy cover, and thinning to 30% canopy cover followed by prescribed burning. The treatment area is overlaid as a black outline.

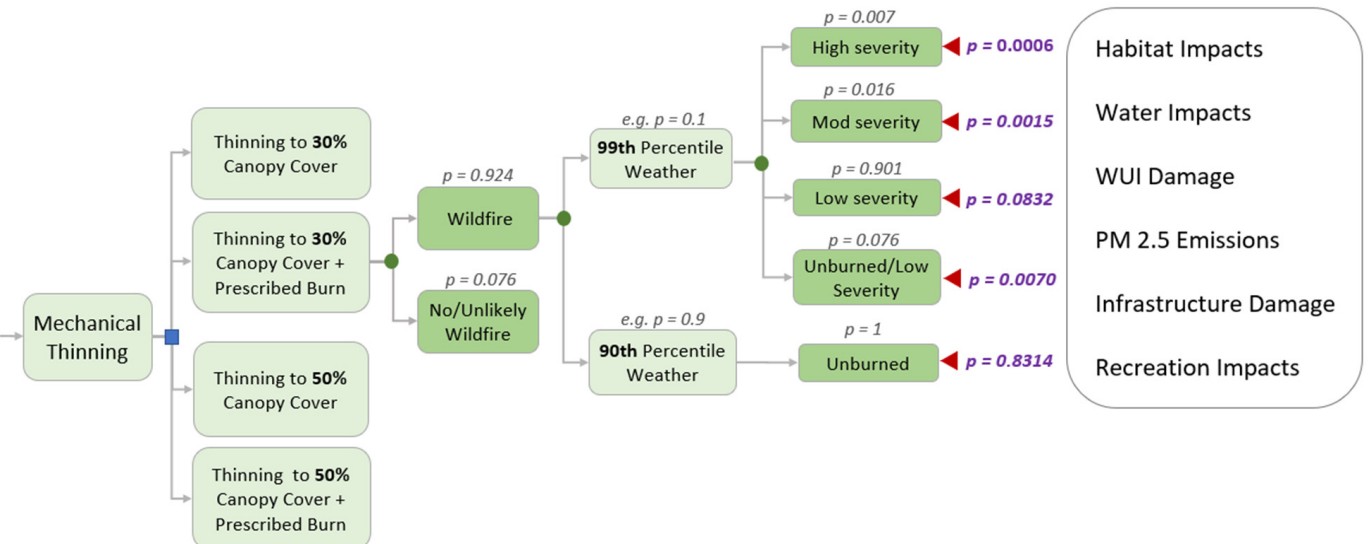

**Figure 9.** Decision analytic representation of mechanical thinning informed by IFTDSS. This figure presents a treatment decision informed by output from Interagency Fuel Treatment Decision Support System (IFTDSS) scenarios. For the purposes of this illustrative decision tree, we simplified burn probability into five possibilities: four wildfire outcomes including with a burn probability of either "Low" (20–40%), "Moderate" (40–60%), "High" (60–80%), or "Highest" (80–100%), and a "No/Unlikely Wildfire" outcome with a burn probability of 0–20%. The probabilities highlighted in purple at each terminal node represent the probability of each possible future outcome occurring conditional on all prior decision and chance nodes to its left.

The QWRA analysis of UWPP landscape suggests that future wildfires are much more likely to result in overall harm than benefit, and, thus, neither proposed treatment alternative is expected to achieve treatment objectives. We offer quantitative confirmation of the concerns expressed by our stakeholders by demonstrating that many of the values at risk that they explicitly identified as being of greatest concern (such as the white-headed woodpecker, NSO, old growth forests, and the WUI) are indeed associated with the highest

levels of vulnerability to the threat of wildfire based on the geospatial configuration of these values on the landscape (see Figure 10).

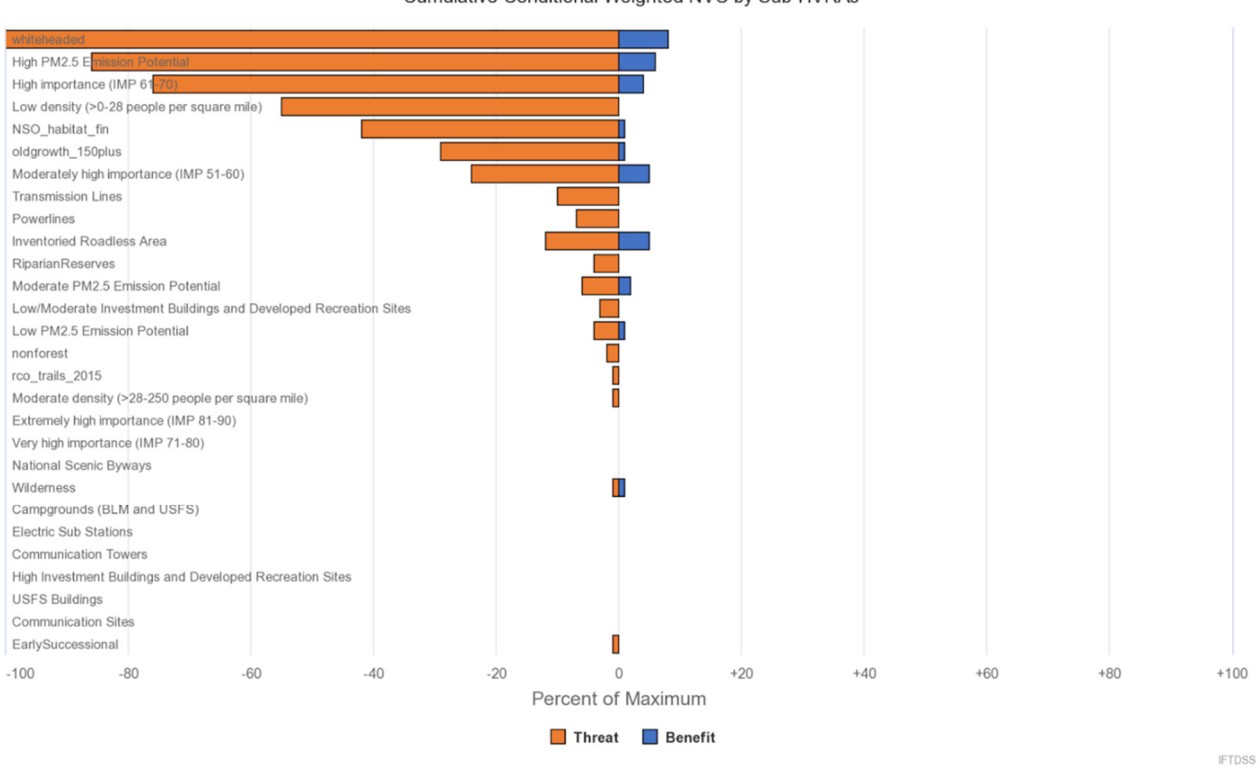

**Figure 10.** Cumulative conditional weighted net value change (NVC) by sub-HVRA. This figure represents expected impacts (threats or benefits) to each HVRA in Upper Wenatchee Pilot Project Alternative 1 after experiencing thinning to 30% canopy cover coupled with prescribed burning (provided by IFTDSS QWRA report). In Appendix G, these results and a corresponding set of results for Alternative 2 are compared.

## 5. Discussion

### 5.1. Multiple Competing Values at Risk

Under warmer and drier summer conditions, the UWPP landscape and its many resources and assets are threatened by the intensifying risk of extreme wildfire. In this setting, there are multiple values at risk competing for policy attention and risk management prioritization. Complex tradeoffs between these values are expressed by our expert stakeholders. The QWRA within IFTDSS confirmed the importance of these competing values across the landscape. Stakeholders remarked that the process of assigning weights and numeric rankings to vital but disparate concepts and values such as ecosystems and recreation was challenging and that they were concerned about oversimplification when expressing relative ranks, given the many competing values and tradeoffs. Future work building upon this methodology will likely also be subject to the challenges of assessing, prioritizing, and placing numeric weights on the values at risk (HVRAs). However, we argue that these challenging and uncomfortable conversations and rankings are critical because prioritization is the foundation upon which tradeoffs in policy- and decision-making processes depend.

Further research in this area may benefit from conducting multiple QWRAs to simulate prioritization of different HVRAs, especially where there are different perspectives among experts about what values are of greatest concern. For example, a faction of our stakeholders expressed interest in exploring and comparing the tradeoffs both from a perspective which considers People and Property to be of the highest importance (as was modeled in our case

study) and separately from a perspective where Habitat is most important (which was a very close second).

### 5.2. Influence of Treatments on Landscape Burn Probability and Conditional Flame Lengths

In this study, we used IFTDSS scenarios to evaluate a range of treatment options and how they influence both overall landscape burn probability and conditional flame lengths. Within UWPP Alternative 1, which prioritized greater fuel treatment areas and less NSO habitat than Alternative 2, our analysis found no appreciable difference in burn probability between thinning alone and thinning combined with prescribed burning. Because treatments are primarily confined to the interior of the study area with minimal treatment in the northern and southern portions of the study area, we expected burn probability across the study area to be similar across scenarios. We do not find substantial differences in burn probability between UWPP Alternatives 1 and 2. In fact, in most instances, landscape burn probability varies only by a fraction of a percent across Alternative 1 and 2; thus, for these measures, the impact of prioritization on fuel treatment versus habitat is found to be relatively modest.

In contrast, conditional flame length predictions vary markedly between treatment scenarios. Given a greater than 90% chance that combined thinning and prescribed burning result in low flame lengths (0–4 feet or 0–1.2 m, Figure 6), managers and firefighters have a range of effective response options available to them (Table 3). By contrast, in the case of thinning alone, there is a near 70% chance of moderate (4–8 feet or 1.2–2.4 m) flame lengths in both Alternatives 1 and 2, i.e., regardless of whether fuel treatment or habitat is prioritized. For both alternatives, thinning combined with prescribed burning is shown as likely to be more effective in reducing fire intensity (conditional flame length) than thinning alone. This finding is consistent with well-established findings in the literature [24,25]. Even though pixel-based outcomes differ markedly by treatment, the overall difference in conditional flame length between Alternatives 1 and 2 is minimal.

In the fire weather scenarios modeled, predicted winds were from the west/southwest direction. Under a more distributed fuel treatment scenario, we might have expected a more noticeable reduction in burn probability as a result of treatment across the landscape, particularly on the other side (east) of prevailing winds [62,63]. However, given that conditional burn probability was already very low across the majority of the landscape prior to treatment (i.e., in the status quo), there was little room for further reduction in BP. The QWRA underscored that large wildfire growth will occur within a relatively rare event but with major potential impacts to the planning area. Record-setting heat events, episodic drought, and summer windstorms are all becoming more common under climate change [2], and landscapes such as the UWPP that have mostly avoided fire in the relatively recent past are increasingly at risk.

Treatment scenarios were based on two alternatives that are actively being considered by the USFS for the UWPP. Landscape evaluations using conditional burn probability and conditional flame length both demonstrated that, as planned, neither alternative is likely to mitigate the probability and impacts of wildfires overall. However, based on pixel outcomes as depicted in Figures 7 and 8, lower conditional flame lengths were observed in scenarios with both thinning and combined thinning and prescribed burning treatment units, with a much higher probability of reduced severity in the latter. Because the majority of the study area is excluded from treatment even in Alternative 1 (which includes a greater share of treated land than Alternative 2 and favors more land reserved for NSO habitat), neither alternative reduced the expected burn probability for untreated areas.

### 5.3. Decision Analytic Frameworks

Decision analytic frameworks, such as decision trees, are recognized as a valuable tool to inform probabilistic outcomes from land management decisions with uncertain outcomes [4]. This study evaluated an improved methodology for utilizing existing wildfire

decision support systems such as IFTDSS to improve risk analysis and evaluate tradeoffs among competing values at risk.

Our novel application of a decision analytic approach offers an example of how fuel treatment scenarios in IFTDSS can be evaluated to help weigh the tradeoffs of impacts between management alternatives. Land managers are increasingly challenged to operate in the face of uncertainty, and decision trees can help mitigate this difficulty by clearly reflecting the relative likelihood of outcomes under different strategies and alternatives. The conditional probabilities derived using our methodology offer decision makers a new tool to holistically consider the impact of planning alternatives on a landscape, values at risk at the landscape scale, and the likelihood of successfully achieving their desired outcomes in selecting a course of treatment. In this study, we presented a simple illustrative decision tree example as a proof of concept, a building block in advance of more complex and integrated landscape-scale evaluations of values at risk. Further development of probabilistic descriptions of uncertainty will be needed before generating a full decision analytic framework capable of representing and evaluating outcomes under multiple treatment scenarios.

## 6. Conclusions

Management decisions for problems as wicked as intensifying risk of wildfire in the 21st century require thorough consideration of competing objectives. This paper utilizes the UWPP case study to propose an enhanced methodology for leveraging existing wildfire decision support systems such as IFTDSS to improve risk analysis through the use of decision analytic frameworks to evaluate tradeoffs more holistically among competing values at risk.

Our results reinforce that IFTDSS offers a helpful platform for understanding integrated risk. Discussions with expert stakeholders, guided by a QWRA, effectively inform the probabilistic outcomes of each treatment alternative proposed for the UWPP and highlight the multiple competing values at risk under current conditions. These results, when incorporated into our decision analytic approach, suggest that the two alternatives within the UWPP offer localized treatment effectiveness, and that treatments that involve combined thinning and prescribed burning are expected to be most effective. However, neither alternative is expected to meaningfully influence predicted fire behavior at the landscape level. In order to achieve substantial reductions in potential wildfire behavior and impacts, particularly in extreme fire weather conditions, a greater proportion of the study area landscape would need to undergo treatment.

More broadly, these results establish that decision analytic approaches—such as decision trees—can functionally inform key probabilistic outcomes for a range of values at risk, which can be used to inform planning decisions. Due to their demonstrated ability to make probabilistic outcomes explicit for challenging land management decisions with uncertain outcomes, such decision analytic approaches hold substantial promise for improving future landscape evaluations.

**Author Contributions:** Conceptualization, A.C.C., S.J.P. and H.K.S.; Methodology, A.C.C., S.J.P. and H.K.S.; Validation, H.K.S., S.J.P. and A.C.C.; Formal Analysis, H.K.S.; Investigation, H.K.S.; Resources, A.C.C. and S.J.P.; Data Curation, H.K.S.; Writing—Original Draft Preparation, H.K.S.; Writing—Review & Editing, H.K.S., A.C.C. and S.J.P.; Visualization, H.K.S.; Supervision, A.C.C. and S.J.P.; Project Administration, H.K.S., S.J.P. and A.C.C.; Funding Acquisition, A.C.C. All authors have read and agreed to the published version of the manuscript.

**Funding:** The authors gratefully acknowledge support from the National Science Foundation Growing Convergence Research Award #2019762.

**Institutional Review Board Statement:** Not applicable.

**Informed Consent Statement:** Informed consent was obtained from all subjects.

**Data Availability Statement:** The data that support this study were obtained from a variety of publicly available sources which are referenced with full citations in the manuscript.

**Acknowledgments:** The authors are grateful to Brian Goldgeier (University of Washington Evans School of Public Policy and Governance) for his technical assistance. We also thank the group of local experts who participated in interviews/workshops and offered vital insights.

**Conflicts of Interest:** The authors declare no conflict of interest.

## Appendix A. Interview Questions

(1)  What is/are your areas of expertise?

- Silviculture & Forestry
- Fire Science/Ecology
- Aquatics
- Wildlife Ecology
- Landscape Ecology
- Other

(2)  Consider the following highly valued resources and assets (HVRAs). Rank the HVRAs according to their importance in terms of the risk wildfire poses to them based on your own understanding and perspective, where 1 is most important.

- Habitat
- People & property
- Air quality & emissions
- Recreation
- Surface drinking water
- Infrastructure (i.e., powerlines, communication towers, etc.)

(3)  Are there any other HVRAs that you feel are not included here that should be?

(4)  How would considering this affect your ranking, if at all?

(5)  Are there any HVRAs that feel like they are equally important? If so, why?

(6)  Let's break each of these highly valued resources and assets (or "primary HVRAs") into sub categories (or "sub-HVRAs") and assign relative importance. Relative importance is a numerical score assigned to an HVRA between 0–10 where 10 is most important that allows us to account for tradeoffs associated with competing objectives.

(7)  Response functions are a measure of susceptibility or resilience of each sub-HVRA to wildfires of varying intensities. Response function can be positive (benefitting from fire) or negative (harmed by fire) between $-100$ and $100$ where $-100$ is total destruction and $100$ is resilience and flourishing, in terms of first order fire effects (direct or indirect immediate consequences of fire, such as tree mortality, soil heating biomass consumption, etc.). We define intensity by flame length ranging from 0 to 12+ feet (0 to 3.7+ meters).

## Appendix B. 90th and 99th Percentile Weather Conditions

Fire weather conditions (90th and 99th percentile scenarios) were defined using RAWS Station 452128 (Viewpoint), which is the station closest to the study area. Weather conditions were determined using data from 2012–2022 for fire season months (June–September). See further specifications below.

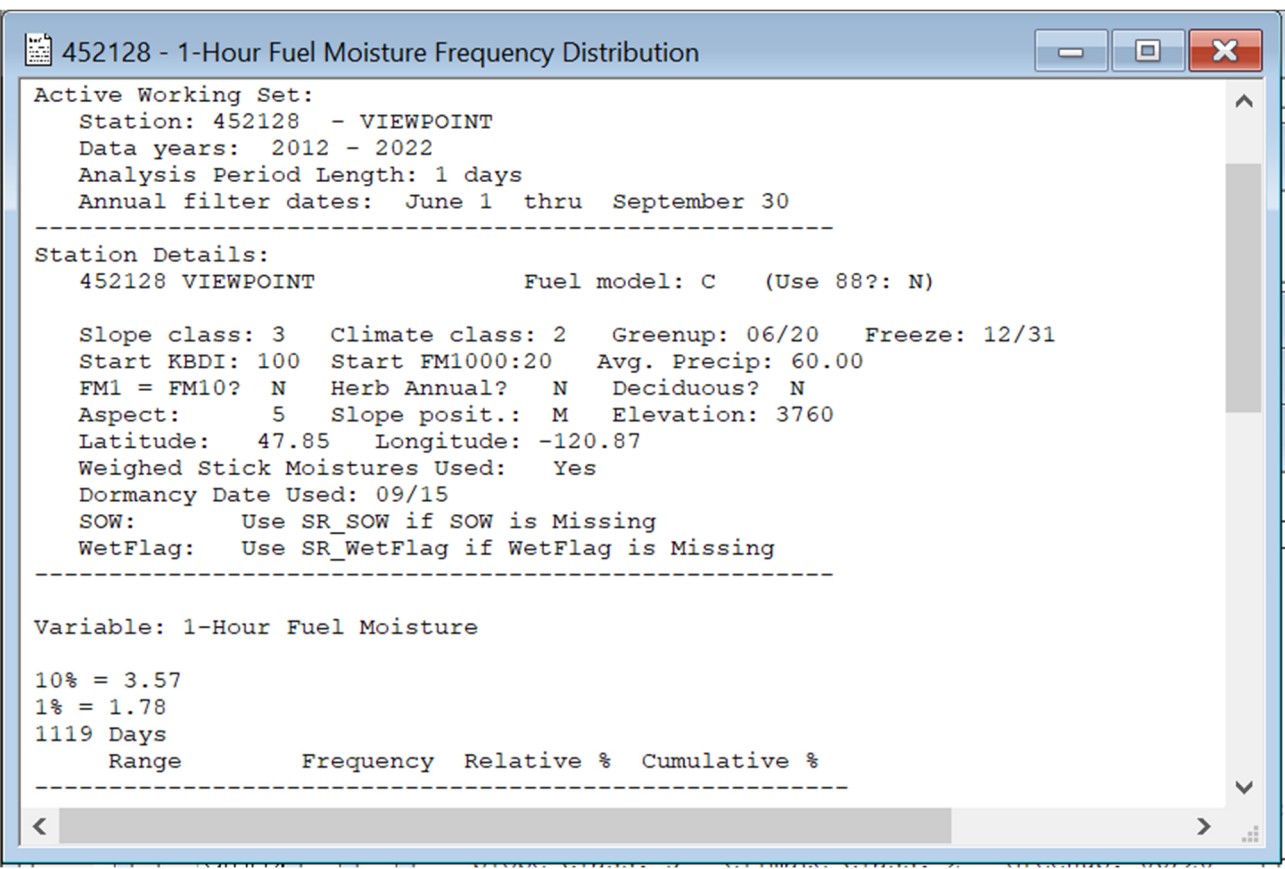

**Figure A1.** Fire weather condition output from RAWS 452128 (Viewpoint) 2012–2022. Specifications that could not be informed by RAWS data were determined based on recommendations from the publicly available IFTDSS documentation or IFTDSS Help Desk (Tables A1 and A2). These specifications include wind subcategory (generate gridded winds), crown fire method, foliar moisture content, fuel moisture conditioning, ignitions, and spotting probability.

For both Tables A1 and A2 (below), selections for which there is not a source cited were determined based on IFTDSS Help Desk recommendations, default selections, and intuition of the researchers. All simulations used the same ignitions. For the first simulation, we selected "Use Random Ignitions", and all simulations thereafter used those points of ignition ("Use Ignitions from a Completed Run").

**Table A1.** 90th percentile fire weather conditions—IFTDSS specifications.

| CATEGORY | SUBCATEGORY | SELECTION 1 | SELECTION 2 | SOURCE |
|---|---|---|---|---|
| **Wind** | **Generate Gridded Winds** | **Wind Speed** | 8 mph | RAWS Station 452128 |
| | | **Wind Direction** | 225° | RAWS Station 452128 |
| **Crown Fire Inputs** | **Crown Fire Method** | Scott/Reinhardt | | |
| | **Foliar Moisture Content** | 100% | | |

**Table A1.** *Cont.*

| CATEGORY | SUBCATEGORY | SELECTION 1 | SELECTION 2 | SOURCE |
|---|---|---|---|---|
| **Initial Fuel Moisture** | **1 h FM** | 4 | | RAWS Station 452128 |
| | **10 h FM** | 7 | | RAWS Station 452128 |
| | **100 h FM** | 8 | | RAWS Station 452128 |
| | **Herb FM** | 60 | | Fuel Characteristics Classification System [64] |
| | **Woody FM** | 90 | | Fuel Characteristics Classification System [64] |
| **Fuel Moisture Conditioning** | **Condition (Select Classified Weather Stream)** | Extreme | | |
| **Ignitions** | | Use ignitions from completed run | | |
| **Simulation Time (Burn Period Length)** | | 12 h | | |
| **Spotting Probability** | | 20 percent | | |

**Table A2.** 99th percentile fire weather conditions—IFTDSS specifications.

| CATEGORY | SUB-CATEGORY | SELECTION 1 | SELECTION 2 | SOURCE |
|---|---|---|---|---|
| **Wind** | **Generate Gridded Winds** | **Wind Speed** | 25 mph | RAWS Station 452128 |
| | | **Wind Direction** | 248° | RAWS Station 452128 |
| **Crown Fire Inputs** | **Crown Fire Method** | Scott/Reinhardt | | |
| | **Foliar Moisture Content** | 100% | | |
| **Initial Fuel Moisture** | **1 h FM** | 2 | | RAWS Station 452128 |
| | **10 h FM** | 3 | | RAWS Station 452128 |
| | **100 h FM** | 4 | | RAWS Station 452128 |
| | **Herb FM** | 30 | | Fuel Characteristics Classification System [64] |
| | **Woody FM** | 60 | | Fuel Characteristics Classification System [64] |

**Table A2.** *Cont.*

| CATEGORY | SUB-CATEGORY | SELECTION 1 | SELECTION 2 | SOURCE |
|---|---|---|---|---|
| **Fuel Moisture Conditioning** | **Condition (Select Classified Weather Stream)** | Extreme | | |
| **Ignitions** | | Use ignitions from completed run | | |
| **Simulation Time (Burn Period Length)** | | 12 h | | |
| **Spotting Probability** | | 20 percent | | |

## Appendix C. Highly Valued Resources and Assets

*Appendix C.1. Habitat*

We specify several sub-HVRAs when defining Habitat: Northern spotted owl (NSO), fish, old growth forest, white-headed woodpecker, nonforest, and other early successional habitat. NSO habitat is represented as mapped in USFS planning documents. Fish habitat is represented using riparian zones. Old growth forests are defined as mature conifer forests (150+ years old), as mapped in 2004 by the Conservation Biology Institute [65]. White-headed woodpecker habitat is represented using the USGS range extent based on 2001 ground condition [66]. Nonforest habitat and early successional habitat are defined based on LANDFIRE 2020 existing vegetation types [67].

*Appendix C.2. People and Property*

People and Property (or "Communities", as described in IFTDSS) are defined in terms of population density using IFTDSS' default reference layer data [68]. This HVRA is meant to estimate wildfire risk and impacts to the WUI. Communities are represented based on residentially developed populated area (RDPA) [69]. RDPA was developed using the LandScan USATM population data [70]. RDPA is summarized into three population density classes: low (>0–28 people per square mile, medium (>28–250 people per square mile), and high density (>250 people per square mile) [71].

*Appendix C.3. Air Quality and Emissions*

Air Quality and Emissions represents the potential for fine particulate matter (PM2.5) from wildfire emissions. These data were constructed by USFS using the First Order Fire Effects model (FOFEM) to predict point source emission potential. Inputs for FOFEM come from LANDFIRE and Forest Inventory Analysis. Emission potential does not include smoke dispersion or impact to populations. The USFS classifies three emission potential classes by dividing the distribution of the data into thirds: low, moderate, and high PM2.5 emission potential [71].

*Appendix C.4. Recreation*

Recreation is represented by specifying the following sub-HVRAs: scenic areas, roadless areas, designated wilderness, and trails. Scenic areas are represented using national scenic byways, as designated by the USFS Geospatial Service and Technology Center, which is a default reference layer of IFTDSS [68]. Roadless areas are also represented using IFTDSS' default reference layers [68], originally shared by the National Parks Service. Designated wilderness is represented using IFTDSS' default reference layers, which combine the best available data from the federal agency responsible for administration of a given wilderness area. Where larger-scale data could not be obtained, the National Wilderness Preservation System map was used as the data source [72]. Federal, state, local, and "other" trails

are represented using the Washington State Recreation and Conservation Office's Trails Database Project [73]. This dataset is preferred over trail data offered by IFTDSS, which did not appear to offer a comprehensive or current map of trails in the UWPP region.

*Appendix C.5. Surface Drinking Water*

Surface Drinking Water is defined by municipal drinking water data derived from the USDA Forest Service Forests to Faucets project [74]. The Surface Drinking Water Importance (IMP) index takes into account water supply, spatial flow through the landscape, and downstream drinking water demand [74]. The IMP index, which ranges from 0 to 100, is mapped to the hydrologic unit 12 (HUC 12 scale). Our study area includes moderately high (IMP 51–60), high (IMP 61–70), and very high importance (IMP 71–80) areas.

*Appendix C.6. Infrastructure*

Infrastructure is defined as the following sub-HVRAs: power lines, communication towers, other communication sites, USFS buildings, electric substations, transmission lines, low/moderate-investment buildings and developed recreation sites, high-investment buildings and developed recreation sites, and campgrounds owned by BLM and USFS [68].

**Appendix D. Burn Probability Distribution**

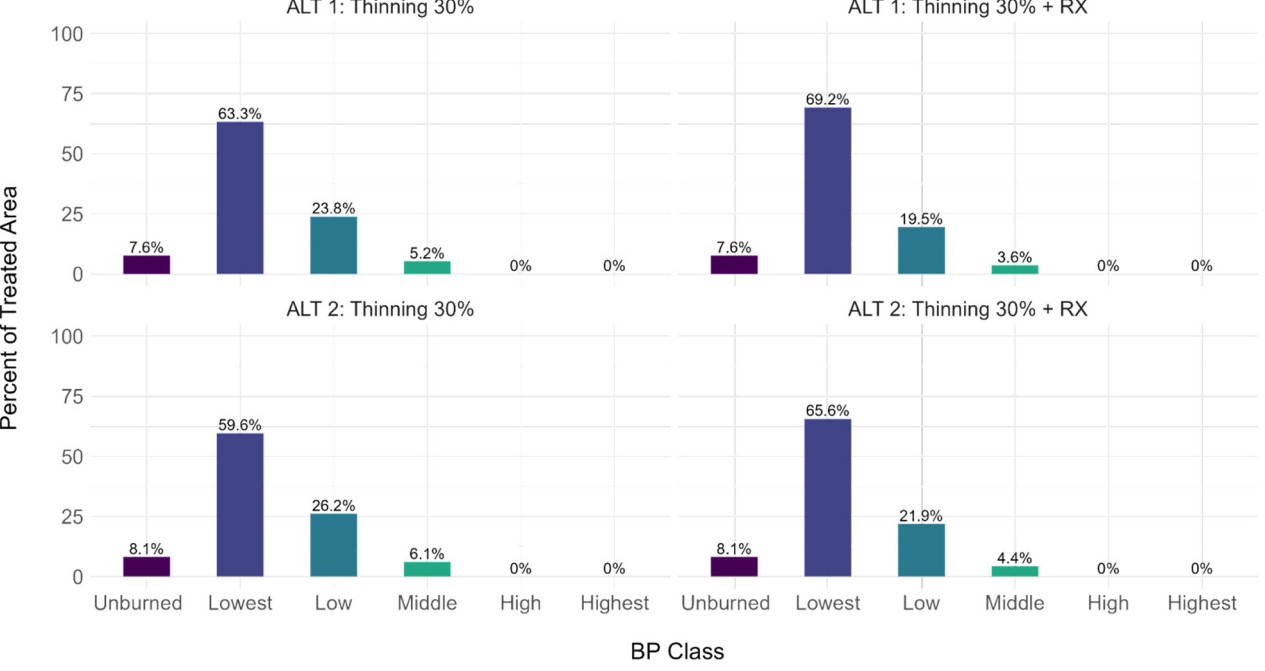

**Figure A2.** This figure depicts the percent of the treated area belonging to each burn probability class for both treatment alternatives subject to 99th percentile fire weather. Burn probability class is defined as follows: "Unburned" implies the share of landscape that was burnable but did not burn, "Lowest" = 0–20% of the analysis maximum, "Low" = 20–40%, "Middle" = 40–60%, "High" = 60–80%, and "Highest" = 80–100%. This figure represents conditional flame length distribution for both treatment alternatives subject to 99th percentile fire weather.

## Appendix E. Burn Probability and Conditional Flame Length Distributions by Forest Type

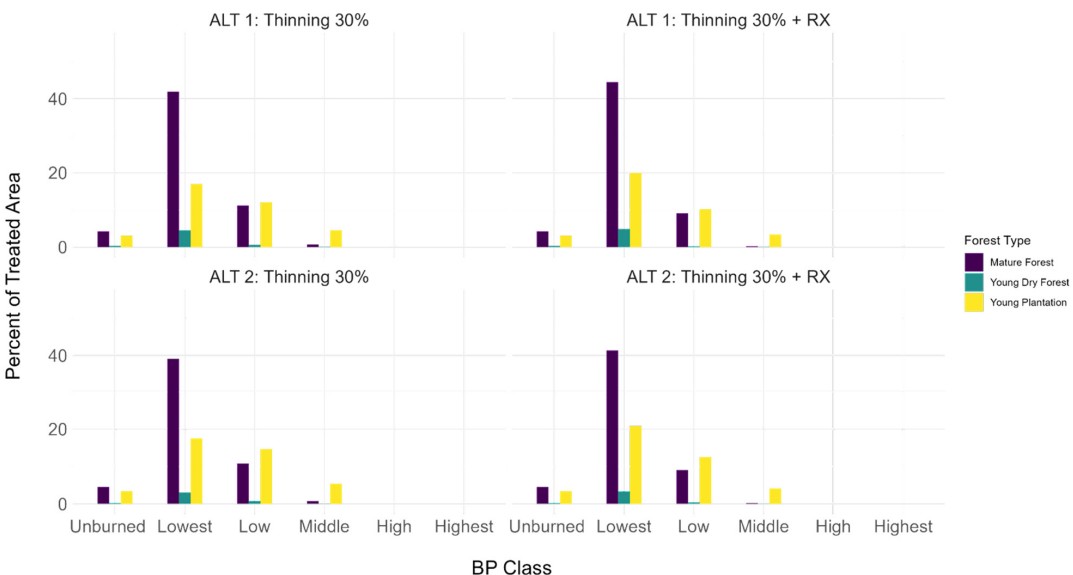

**Figure A3.** Burn probability distribution. This figure presents the percent of the treated area belonging to each burn probability class for both treatment alternatives subject to 99th percentile fire weather, disaggregated by forest type. Burn probability class is defined as follows: "Unburned" implies the share of landscape that was burnable but did not burn, "Lowest" = 0–20% of the analysis maximum, "Low" = 20–40%, "Middle" = 40–60%, "High" = 60–80%, and "Highest" = 80–100%. This figure represents conditional flame length distribution for both treatment alternatives subject to 99th percentile fire weather.

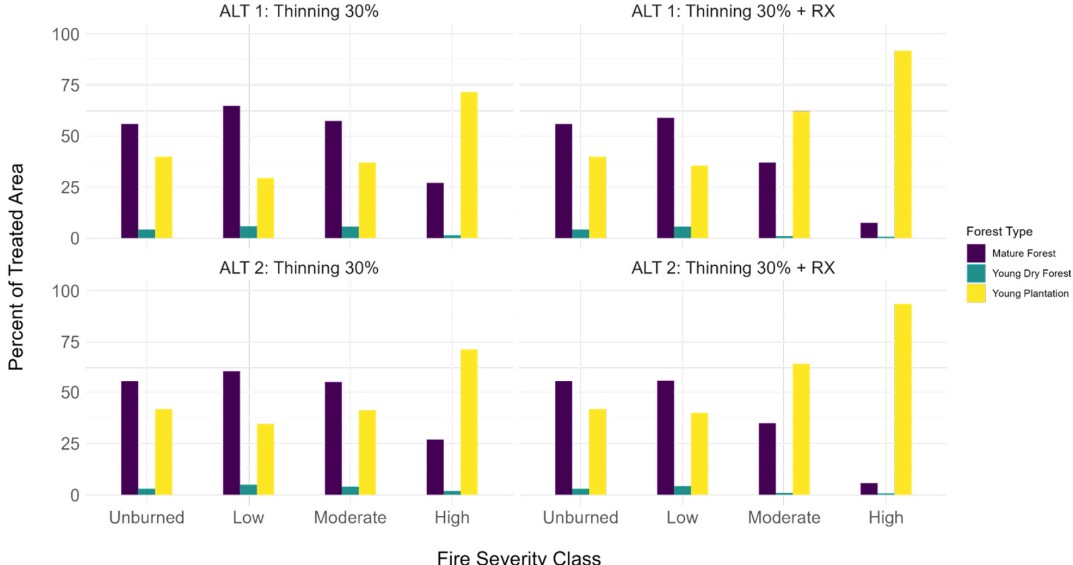

**Figure A4.** Conditional flame length distribution. This figure depicts the percent of the treated area belonging to each conditional flame length class for both treatment alternatives subject to 99th percentile fire weather, disaggregated by forest type. Fire severity class based on conditional flame length is defined as follows: "Unburned" implies the share of landscape that was burnable but did not burn, "Low" = 0–4 feet (0–1.2 m), "Moderate" = 4–8 feet (1.2–2.4 m), and "High" = 8–12+ feet (2.4–3.7+ meters). Conditional flame length distribution for both treatment alternatives are represented subject to 99th percentile fire weather.

## Appendix F. Probability Estimates for Burn Probability and Conditional Flame Length

**Table A3.** Burn probabilities—99th percentile fire weather.

| Alternative | Scenario | Class | Probability |
|---|---|---|---|
| Alternative 1 | Thinning 30% | Unburned | 7.62% |
| Alternative 1 | Thinning 30% + RX | Unburned | 7.62% |
| Alternative 1 | Thinning 30% | Lowest | 63.31% |
| Alternative 1 | Thinning 30% + RX | Lowest | 69.25% |
| Alternative 1 | Thinning 30% | Low | 23.83% |
| Alternative 1 | Thinning 30% + RX | Low | 19.52% |
| Alternative 1 | Thinning 30% | Middle | 5.24% |
| Alternative 1 | Thinning 30% + RX | Middle | 3.62% |
| Alternative 1 | Thinning 30% | High | 0% |
| Alternative 1 | Thinning 30% + RX | High | 0% |
| Alternative 1 | Thinning 30% | Highest | 0% |
| Alternative 1 | Thinning 30% + RX | Highest | 0% |
| Alternative 2 | Thinning 30% | Unburned | 8.14% |
| Alternative 2 | Thinning 30% + RX | Unburned | 8.13% |
| Alternative 2 | Thinning 30% | Lowest | 59.59% |
| Alternative 2 | Thinning 30% + RX | Lowest | 65.57% |
| Alternative 2 | Thinning 30% | Low | 26.16% |
| Alternative 2 | Thinning 30% + RX | Low | 21.94% |
| Alternative 2 | Thinning 30% | Middle | 6.11% |
| Alternative 2 | Thinning 30% + RX | Middle | 4.35% |
| Alternative 2 | Thinning 30% | High | 0% |
| Alternative 2 | Thinning 30% + RX | High | 0% |
| Alternative 2 | Thinning 30% | Highest | 0% |
| Alternative 2 | Thinning 30% + RX | Highest | 0% |

**Table A4.** Conditional flame length probabilities—99th percentile fire weather.

| Alternative | Scenario | Class | Probability |
|---|---|---|---|
| Alternative 1 | Thinning 30% | Unburned | 7.62% |
| Alternative 1 | Thinning 30% + RX | Unburned | 7.62% |
| Alternative 1 | Thinning 30% | Low | 20.18% |
| Alternative 1 | Thinning 30% + RX | Low | 90.10% |
| Alternative 1 | Thinning 30% | Moderate | 69.58% |
| Alternative 1 | Thinning 30% + RX | Moderate | 1.61% |
| Alternative 1 | Thinning 30% | High | 2.62% |
| Alternative 1 | Thinning 30% + RX | High | 0.67% |
| Alternative 2 | Thinning 30% | Unburned | 8.13% |
| Alternative 2 | Thinning 30% + RX | Unburned | 8.14% |
| Alternative 2 | Thinning 30% | Low | 19.06% |
| Alternative 2 | Thinning 30% + RX | Low | 89.32% |
| Alternative 2 | Thinning 30% | Moderate | 69.68% |
| Alternative 2 | Thinning 30% + RX | Moderate | 1.79% |
| Alternative 2 | Thinning 30% | High | 3.12% |
| Alternative 2 | Thinning 30% + RX | High | 0.76% |

## Appendix G. Comparison of Conditional Net Value Change for HVRAs across Alternatives

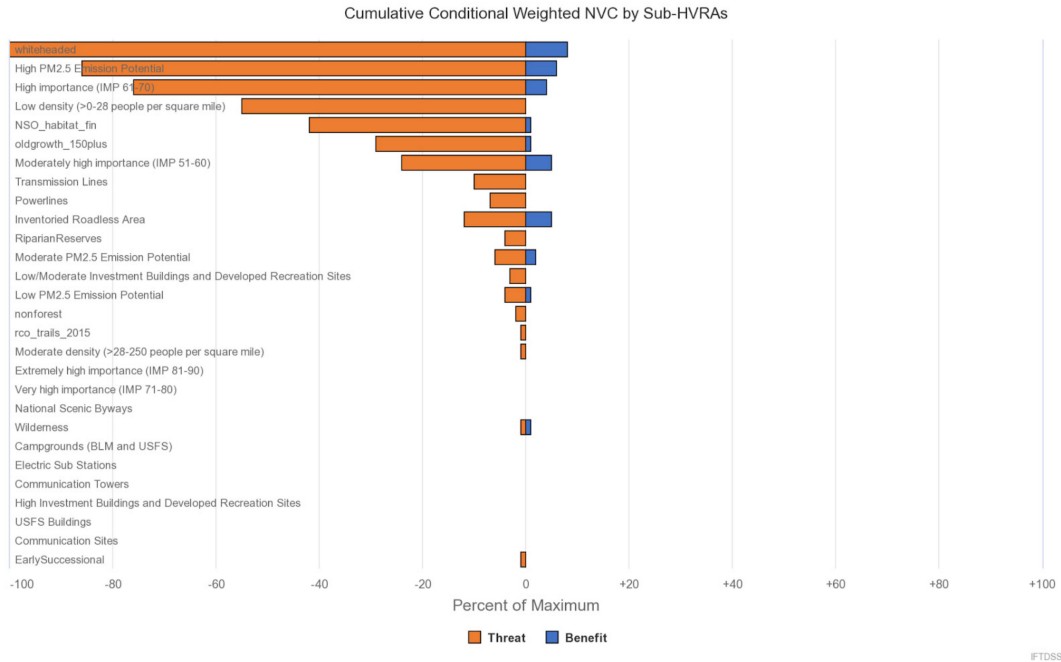

**Figure A5.** Cumulative conditional weighted net value change (NVC) by sub-HVRA for UWPP Alternative 1, after thinning to 30% followed by prescribed burning, subject to 99th percentile fire weather conditions.

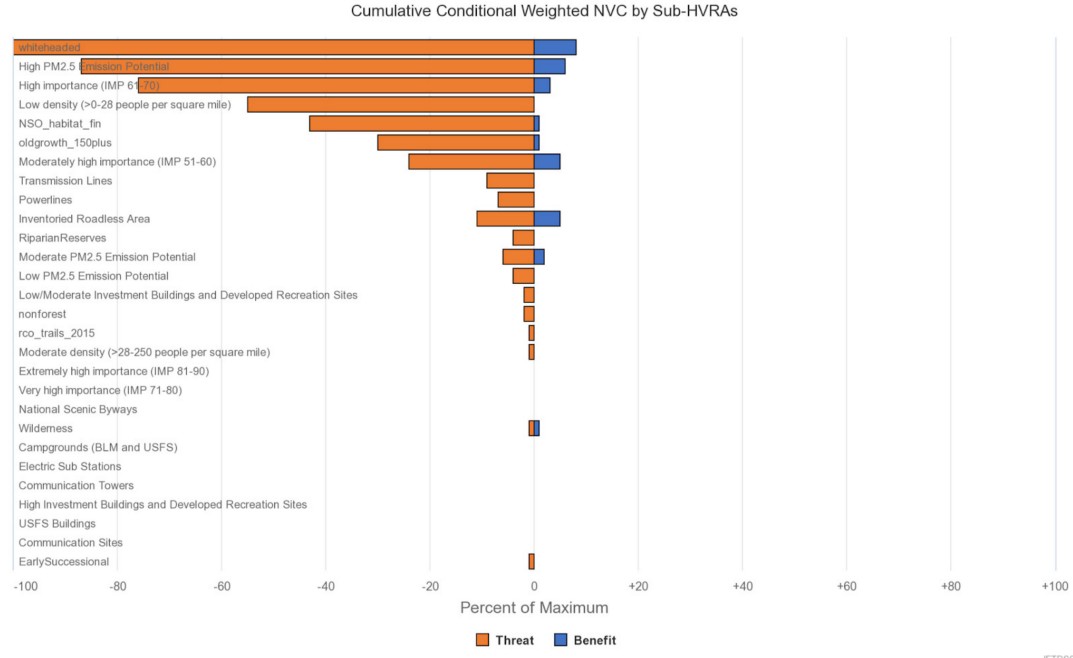

**Figure A6.** Cumulative conditional weighted net value change (NVC) by sub-HVRA for UWPP Alternative 2, after thinning to 30% followed by prescribed burning, subject to 99th percentile fire weather conditions.

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
