# Peer review of "Decision Support for Landscapes with High Fire Hazard and Competing Values at Risk: The Upper Wenatchee Pilot Project"

_fire, doi:10.3390/fire7030077_

Round 1

Reviewer 1 Report

Comments and Suggestions for Authors

Dear Autors,

We are sending a document with suggested changes. A few comments concern the research methodology and the layout of diagrams.

The layout of diagrams should be discussed with the editors.

Author Response

Responses to Reviewer 1.

Line 11 comment #1: Thank you for pointing this out; this text has been removed

Line 11 comment #2: Thank you for your suggestion. We have altered the language to be more specific about the impact that climate change has had on wildfires.

Line 90: Thank you, this comment has been addressed and is highlighted in the resubmission.

Line 143: Thank you for this comment. Figure titles were lost in the original transmission. This has been addressed and all titles have been restored.

Line 164: Thank you for this comment. Titles were lost in translation. This has been addressed.

Lines 167-168: Thank you for your concern. We worked closely with stakeholders active in the management activities for this area and with the Upper Wenatchee Pilot Project itself.  Thus we are confident in the legality and applicability of these alternatives in LSRs.

Line 174: Thanks for the suggestion. It has been implemented accordingly and is highlighted in the resubmission.

Line 198: The illegible text has been addressed, thank you. We experimented with modifying the layout of the maps and feel that it is easier to detect differences and make comparisons across the two when laid out horizontally, but we defer to the editors to determine the most suitable layout. To make it easier for the layout and editorial staff to decide on orientation we have separated the panels from each other in the resubmission and the legend to enhance size and readability. If the editors opt to place them vertically, we encourage them to change the language in the caption accordingly (“Right” and “Left” panels to “Top” and “Bottom” panels).

Line 207: Thank you for drawing attention to this. The appendices were submitted with the original document but not inserted accordingly in later communications. This has been addressed and all appendices have been restored to the resubmission and placed after the References.

Line 232: We appreciate your comment. These features of a fire-prone landscape are indeed important. We would note however that the decision under consideration by managers is not what to do with a barren landscape, but rather how to mitigate risk on the landscape “as is”.  The landscape “as is” includes existing infrastructure such as service roads and fire lookouts. For the managers we seek to inform, it is beyond their scope to consider risk management alternatives for a landscape that has been significantly altered in these ways. 

Line 235: We thank Reviewer 1 for this comment.  We prefer to keep decision analytic frameworks in a left-to-right flow for ease of comprehension and consistency with similar analytic approaches found in the literature.

Line 242.: Figure captions have been inserted. Thank you. We are opting to keep the “RX” abbreviation to maintain readability and conserve space; the term “RX” is commonly understood in the wildland fire community to mean prescribed fire.

Line 285: Thank you. Appendices have been restored.

Line 314: We appreciate your concern. We have amended the manuscript accordingly as highlighted in the resubmission file.

Line 347: We appreciate your concern. We are not in a position to make this change because this figure is a direct output of the IFTDSS software which is a core element of our approach. Including this output in the IFTDSS format improves the direct usability and applicability for practitioners and professionals in the field.

Reviewer 2 Report

Comments and Suggestions for Authors

My apologies for the long delay in providing this review.

I felt that this paper read more like a technical or working group report than a scientific study, and also that it assumed a high degree of familiarity with North American fire management protocols and tools such as the Interagency Fuel Treatment Decision Support System (IFTDSS). I also found it difficult to follow, or to know what the reader was meant to take from it.

The paper makes frequent references to a "decision tree" but doesn’t clearly define exactly what this is. Given my statistical background I initially assumed it to be some form of statistical model, e.g. a Bayesian network or similar, but this doesn’t seem to be the case. I then wondered if the term simply referred to the dichotomous tree depicted in figures 4 and 5. Or perhaps it is some module within the IFTDSS? By the end of the paper I still didn’t know.

More generally, I was left in the dark about exactly what this study was trying to demonstrate and what, if anything, was novel or innovative about the methods used. It seemed that there was a largely qualitative workshop process, with options informed (in some way that was not clear) by the results of fire simulations. Workshop participants nominated (perhaps reluctantly?) that the key concern was people and property. Finally, an analysis (the details of which are not explained sufficiently to comment on) found that neither of the management options under consideration were acceptable. What is the reader to take from this?

In comparison, consider another recent study of expert-informed fire management planning in North America (note: I am not an author on this paper and have no association with the authors):

Nepal et al. 2023. Wildfire Risk Assessment for Strategic Forest Management in the Southern United States: A Bayesian Network Modeling Approach. Land 12, 2172

This paper gives a full account of the methods used to consider competing aims in the light of knowledge and process uncertainty, and present outputs in an interesting and understandable manner that is relevant to an international readership.

I am sorry, but I feel that the manuscript is simply not suitable for publication in Fire. Perhaps a local publication would be more appropriate.

Given my major concerns I am not providing specific comments on the text.

Author Response

We have edited the manuscript in response to specific reviewer comments and also these general comments from Reviewer #2 and feel that we have improved clarity of implications for readers. We are confident that the specificity of our case study to the western North American context is useful to our intended audience, as confirmed by stakeholders, and believe that the broader implications can serve a more global audience.

We have further refined and expanded the description and content related to our decision analytic approach (see lines 49-53 and cited literature) and the decision trees that are used to organize and convey information about management alternatives available to decision makers, uncertainties, tradeoffs and consequences.  We appreciate the suggestion of applying a network analysis or an influence diagram (i.e., other decision analytic approaches) while finding that a decision tree structure is well suited to our purpose and to the level of information available in our analysis.

Further, we have added clarifying language to our conclusions where we note that the two alternatives examined offer localized treatment effectiveness, and that in general treatments that involve thinning and prescribed burning are expected to be most effective.  We add that in order to achieve substantial reductions in potential wildfire behavior and impacts, particularly during extreme fire weather conditions, a greater proportion of the landscape would need to undergo treatment. See also responses to Reviewer #4.

Reviewer 3 Report

Comments and Suggestions for Authors

This study presents a robust approach to address the escalating threat of intensified wildfires resulting from climate change. Through a clear background introduction, the article effectively outlines the challenges posed by wildfire risks to land managers, highlighting the complexities of decision-making and the need for balancing various interests. The proposal to utilize a decision analytic framework to complement decision support systems is a promising idea and holds positive implications for tackling this multifaceted issue.

Methodologically, the research employs a quantitative wildfire risk assessment, translating results into probabilistic descriptions of wildfire occurrence and intensity. The construction of a decision tree to explicitly evaluate tradeoffs among treatment alternative outcomes demonstrates the feasibility and innovation of the study's approach. Active engagement with stakeholders enhances the practical applicability of the research.

Concerning the study's findings, the authors clearly present the impact of two alternative treatments on wildfire risk. The conclusion that neither alternative significantly minimizes the impact of wildfires on the landscape effectively communicates the research outcomes. This conclusion, backed by the quantitative approach and stakeholder involvement, underscores the study's significance in addressing the complexities of wildfire risk management.

The study provides a valuable contribution to the field by proposing a systematic approach to address the challenges posed by climate-induced wildfires. The well-defined methodology and clear presentation of results enhance the overall quality and applicability of the research. Before the publication of the manuscript, there are some issues that need clarification.

It is recommended to modify the title to: Decision Support for Landscapes with High Fire Hazard and Competing Values at Risk.

Introduction section lacks depth, making it challenging to engage readers with the story and fails to adequately convey the significance of your research topic. Consider adding more details and context to make the narrative more compelling and highlight the necessity of your study.

Many figures are unclear, and the content in the images is difficult to discern for an unknown reason. Consider providing clearer images or improving the resolution to enhance the readability of the figures.

Are there additional supporting materials for the interviews? If so, please provide them as attachments.

3.3. Decision Analytic Frameworks: Due to the inability to discern the text in the figures (Figure 4), it is challenging to comprehend this section. It is suggested to further refine the methods for clarity.

Author Response

Responses to Reviewer 3:

  1. It is recommended to modify the title to: Decision Support for Landscapes with High Fire Hazard and Competing Values at Risk.

Thank you for this suggestion. The authors would prefer to keep the title as is, as it indicates that this research is a case study.

  1. Introduction section lacks depth, making it challenging to engage readers with the story and fails to adequately convey the significance of your research topic. Consider adding more details and context to make the narrative more compelling and highlight the necessity of your study.

We have added content to the introduction while also relying on the substantial body of literature on this topic to provide detailed information about the challenge that wildfire presents.

  1. Many figures are unclear, and the content in the images is difficult to discern for an unknown reason. Consider providing clearer images or improving the resolution to enhance the readability of the figures.

Reviewer 1 offered several specific concerns in this area that have been addressed. Specifically, the legend on Figures 7 & 8 has been clarified, superfluous text on Figure 3 was eliminated and the legend was magnified for easier readability, and higher resolution versions of Figures 4, 5 and 9 have been inserted.

  1. Are there additional supporting materials for the interviews? If so, please provide them as attachments.

Interview questions have been inserted in Appendix 1. Thank you for your suggestion.

  1. “3.3. Decision Analytic Frameworks”: Due to the inability to discern the text in the figures (Figure 4), it is challenging to comprehend this section. It is suggested to further refine the methods for clarity.

Thank you, I have improved clarity on all figures and included higher resolution versions. I attempted to clarify language surrounding the use of the Landscape Editing function (278-281). I further simplify the language surrounding the derivation and use of percentages to inform our decision tree in lines 352-359.

Reviewer 4 Report

Comments and Suggestions for Authors

This is a piece of fine work. I enjoyed reading it, and I propose acceptance in its current form. Even if one can trace minor issues of questioning or disagreement here and there, my opinion is that they are insignificant when examined in the overall context and content of this contribution.

Author Response

Thank you for your recommendation for acceptance. We have incorporated several changes as suggested by other reviewers and hope that they have enhanced the piece to your liking!

Reviewer 5 Report

Comments and Suggestions for Authors

Considering that the impactcs of anthropogenic climate change on fire, weather extremes and fire season lenght are changing in many parts of the globe, and that this changes can have significant impacts on, e.g., people & property, infrastructure, GHG emissions, changes on natural habitat  etc... the study is highly timely.

Specific comments:

Lines 11 and 12: Does fire season lenght, incresead in western US in past years? Iff so, it is an important information in the sentence, and reflect additional  chalenges for land managers 

Line 125: UWPP = Upper Wenatchee Pilot Project (please explain to readers, there is no mention for UWPP meaning along the manuscript).

Figure 3 Legend difficult to read evean with zoom

Figure 4 Suggestion: Vertical position would better to read

Explain the meanning of Thinning + RX

Figure 7 color pallete could be improved.

My main question is related to abstract and conclusion, one key finding "that treatments that involved thinning and prescribed burning were the most effective." this important result wasn't present in abstract section. The main message at his point to readers was "We find that neither alternative is more likely to significantly minimize the risk of wildfire  impacts on this landscape"

Author Response

Responses to Reviewer 5:

  1. Lines 11 and 12: Has fire season length increased in western US in past years? Iff so, it is an important information in the sentence, and reflect additional  challenges for land managers 

Thank you for your point of clarification. We have clarified the language in this section to more specifically and accurately reflect the changes to wildfire activity that climate change influences (“a strong contributing factor in the lengthening and intensification of wildfire season”), which we substantiate with citations [4,5,6] in the introductory paragraphs to follow.

  1. Line 125: UWPP = Upper Wenatchee Pilot Project (please explain to readers, there is no mention for UWPP meaning along the manuscript).

Thank you for your attention to detail. This has been clarified.

  1. Figure 3 Legend difficult to read even with zoom

This has been addressed, thank you.

  1. Figure 4 Suggestion: Vertical position would better to read

Thank you for this thoughtful suggestion. We experimented with modifying the layout of the maps and feel that it is easier to detect differences and make comparisons across the two when laid out horizontally, but we defer to the editors to determine the most suitable layout. We did, however, separate the panels from each other and the legend to enhance size and readability. If the editors opt to place them vertically, we encourage them to change the language in the caption accordingly (“Right” and “Left” panels to “Top” and “Bottom” panels).

  1. Explain the meaning of Thinning + RX

This has been addressed in Figure 4’s caption. Thank you.

  1. Figure 7 color palette could be improved.

We appreciate your comment. We have opted not to edit this color palette as Figures 7 and 8 are direct outputs from IFTDSS. We believe including the raw outputs will improve applicability of our research to future work of readers, fire managers, and other users of IFTDSS.

  1. My main question is related to abstract and conclusion, one key finding "that treatments that involved thinning and prescribed burning were the most effective." this important result wasn't present in abstract section. The main message at his point to readers was "We find that neither alternative is more likely to significantly minimize the risk of wildfire  impacts on this landscape"

This is an excellent point for further clarification. I have addressed this in the abstract (lines 22-27) and conclusion and reformatted the final paragraphs to clarify that our results are interesting in two ways:

  • For informing the case study specifically: At the landscape level - neither alternative effectively minimizes risk/impacts of wildfire (described in lines 560-563). However, based on pixel-level outcomes (i.e. Figures 7 & 8), the conditional flame lengths were technically lower in both the thinning alone, and the thinning + prescribed burning treatment areas. Further, there was a higher probability of reduced fire severity in thinning + RX treatment areas (558-559).
  • For methodological advancements: broadly, decision analytic approaches are useful for clarifying and making explicit the impacts of challenging tradeoffs (lines 564-570).

Round 2

Reviewer 1 Report

Comments and Suggestions for Authors

I am accept this material.

Reviewer 3 Report

Comments and Suggestions for Authors

The authores almost addressed all my concerns, and this manuscript could be accepted in present form.